# Use of Bioproducts Derived from Mixed Microbial Cultures Grown with Crude Glycerol to Protect Recycled Concrete Surfaces

**DOI:** 10.3390/ma14082057

**Published:** 2021-04-19

**Authors:** Lorena Serrano-González, Daniel Merino-Maldonado, Manuel Ignacio Guerra-Romero, Julia María Morán-del Pozo, Paulo Costa Lemos, Alice Santos Pereira, Paulina Faria, Julia García-González, Andrés Juan-Valdés

**Affiliations:** 1Department of Engineering and Agricultural Sciences, School of Agricultural and Forest Engineering, University of León, Av. De Portugal 41, 24071 Leon, Spain; dmerm@unileon.es (D.M.-M.); ignacio.guerra@unileon.es (M.I.G.-R.); julia.moran@unileon.es (J.M.M.-d.P.); julia.garcia@unileon.es (J.G.-G.); andres.juan@unileon.es (A.J.-V.); 2Associated Laboratory for Green Chemistry-Chemistry and Technology Network (LAQV-REQUIMTE), Department of Chemistry, NOVA School of Science and Technology (FCT NOVA), NOVA University of Lisbon, 2829-516 Caparica, Portugal; pac@fct.unl.pt; 3Applied Molecular Biosciences Unit (UCIBIO), Department of Chemistry, NOVA School of Science and Technology (FCT NOVA), NOVA University of Lisbon, 2829-516 Caparica, Portugal; alice.pereira@fct.unl.pt; 4Civil Engineering Research and Innovation for Sustainability (CERIS), Department of Civil Engineering, NOVA School of Science and Technology (FCT NOVA), NOVA University of Lisbon, 2829-516 Caparica, Portugal; paulina.faria@fct.unl.pt

**Keywords:** construction and demolition waste (CDW), glycerol, recycled concrete, surface treatment, PHA-producing MMC, waterproof

## Abstract

The large increase in the world population has resulted in a very large amount of construction waste, as well as a large amount of waste glycerol from transesterification reactions of acyl glycerides from oils and fats, in particular from the production of biodiesel. Only a limited percentage of these two residues are recycled, which generates a large management problem worldwide. For that reason, in this study, we used crude glycerol as a carbon source to cultivate polyhydroxyalkanoates (PHA)-producing mixed microbial cultures (MMC). Two bioproducts derived from these cultures were applied on the surface of concrete with recycled aggregate to create a protective layer. To evaluate the effect of the treatments, tests of water absorption by capillarity and under low pressure with Karsten tubes were performed. Furthermore, SEM-EDS analysis showed the physical barrier caused by biotreatments that produced a reduction on capillarity water absorption of up to 20% and improved the impermeability of recycled concrete against the penetration of water under pressure up to 2.7 times relative to the reference. Therefore, this bioproduct shown to be a promising treatment to protect against penetration of water to concrete surfaces increasing its durability and useful life.

## 1. Introduction

Global demographic problems, with uncontrollable growth of population, greatly increase the demand for energy [1] and construction materials [2] over the past decade, combining the need to make the latter sustainable into a major global challenge nowadays.

Increase in construction activities has led to a rapid depletion of natural resources, particularly aggregates which are utilized in the production of mortars and concrete [3]. The depletion grow is doubling annually, according to a report by Transparency Market Research, reaching 2.2 billion tons by 2025 [4]. At the same time, there is an immense amount of construction and demolition waste, which in turns generates an important management problem that affects everyone.

All waste materials and by-products should be recycled and reintroduced in the productive chain for the purpose of preserving the environment. One of the most efficient strategies employed in addressing problems related to solid wastes consists on recovering usable materials from waste and use them instead of raw materials [2].

The use of recycled aggregates contributes to decrease the amount of construction and demolition waste produced, but the main problem with the use of recycled aggregates is that as numerous studies have shown [5,6,7,8,9,10], this type of aggregates have lower performance compared to natural aggregate. They present higher water absorption capacity and porosity, lower density, and higher Los Angeles abrasion capacity, which causes a decrease in the properties of the concrete with recycled aggregate. For this reason, it is necessary to enhance the properties and extend the service life of all types of concrete and specially the recycled ones. High-quality and durable concrete is required to reduce the rapid deterioration of concrete structures in severe conditions. The decline of concrete durability is affected by factors such as carbonation, alkali-silica reaction, sulphate attack, freeze-thaw attack, and chloride penetration [11]. All of them are directly or indirectly related to the permeability of the concrete. Water not only acts as the main agent causing the deterioration of concrete, but also as the transport medium for aggressive substances like sulfate or chloride ions [12]. Therefore, concrete water permeability is closely related to the durability of structures [13].

Biofuels are an alternative renewable fuel developed in order to overcome the world energy crisis caused by the exhaustive use of fossil resources [14]. Their worldwide production has grown exponentially in the last decade, in particularly biodiesel. Biodiesel has demonstrated to be a great alternative combustion fuel compared to petroleum diesel, and for this reason its annual global production increased from 533 MB/d in 2015 to 805 MB/d in 2019 [15]. 

Crude glycerol (CG) is the main biodiesel by-product, formed during the trans-esterification reaction for biodiesel production [16]. Considered a waste stream, about 10% of the weight of biodiesel is generated as glycerol [17]. The intensification of the use of biodiesel, resulted in a significant glycerol production, which creates a problem of oversupply and leads to a declining glycerol market price [18]. Those large amounts of CG become a major environmental issue due to the high cost of purification for most fine glycerol applications and also problems related to its landfilling [19]. Since glycerol has low solubility in hydrocarbons, high viscosity, and thermal instability at high temperature, it cannot be added to conventional fuel, discouraging its application as an additive in the combustion engine [20]. Further utilizations of CG were proposed such as its utilization as a carbon source for mixed microbial culture (MMC) in polyhydroxyalkanoates (PHAs) production [14].

Polyhydroxyalkanoates, biopolyesters produced by a vast array of prokaryotic Bacteria and Archaea [21], are eco-friendly and renewable materials [22]. The microorganisms are able to produce PHA can do so during their grown phase [23,24] or in response to various stress conditions, for example with an excess of carbon source and limited concentrations of phosphate, nitrogen, sulfur, or oxygen [25]. PHAs are generally classified into three different classes according to the number of carbons in their monomers: short-chain length, scl-PHA (3–5 carbons), medium-chain length (6–16 carbons), and long-chain length (more than 17 carbons) [26,27]. These polymers can be both thermoplastic or elastomeric, being the simplest one poly(3-hydroxybutyrate) (P(3HB)) [28]. This homopolymer is brittle and present high crystallinity, which limits its applications [29]. The presence of a second monomer, in order to obtain a more elastic and flexible copolymer, expand the number of utilizations of those polymers [30]. Due to PHAs lack of water sorption and diffusion, low porosity and wettability [31], they are considered materials with great potential for protection of concrete surface against harmful environmental agents.

This study proposes a new alternative to control both management issues, relying on the utilization of CG as carbon source for the production of bioproducts derived from polyhydroxyalkanoates-producing MMC, using these bioproducts applied on the surface of recycled concrete. The capacity of the bioproducts to protect the surface by creating a physical barrier for water absorption was analyzed by a combination of tests and microscopic observation under scanning electron microscope (SEM) and energy dispersive spectrometry (EDS). The different tests and analyses carried out are intended to prove that treatment with the biopolymers avoided the penetration of water and, therefore, by decreasing of the multi-scale hydrophilicity inherent to concrete can extend its durability. The ultimate aim of this study is to evaluate this novel treatment to improve the durability of recycled aggregate concrete, in an environmentally friendly way and also to contribute to the management of two very large wastes worldwide, namely glycerol and construction waste.

## 2. Materials and Methods

### 2.1. Materials

#### 2.1.1. Concrete

Concrete samples were produced with the composition listed in Table 1. Concrete components included Portland cement 42.5 N/SR CEM III/A, natural river sand (0/4 mm) and, as coarse aggregate (4/16 mm), a mix of 50% blend of natural siliceous gravel and mixed recycled aggregate (MRA), from a construction and demolition waste management plant of TEC-REC, Tecnología y Reciclado S.L. located in Madrid. The composition of the MRA is presented in Table 2. All aggregates were characterized to ensure that they meet the requirements established by the Spanish structural concrete code EHE-08 [32] and European standard EN 12620 + A1 [33]. The recycled concrete had a water/cement ratio of 0.5 and a target strength of 25 MPa, which has previously been evaluated by García-González et al. [34]. 

To counteract the higher water sorptivity associated with recycled aggregate due to the presence of bound mortar on the surface and the nature of the masonry/fired clay fraction comprising, the MRA was pre-soaked in keeping with industry practice for manufacturing recycled concrete suitable for applications not requiring high mechanical strength [35]. 

Two types of samples were prepared: a 400 mm × 100 mm × 100 mm prismatic sample for the Karsten tube test and ten 100 mm × 100 mm × 100 mm cubes for the determination of capillary water absorption. Steel molds were used for casting of samples. The compaction of samples was carried out in three layers using a poker vibrator. Samples were flattened with a smooth steel-trowel then covered with plastic to prevent early evaporation. Specimens were stripped from their molds after 24 h and placed in a curing room at 20 ± 2 °C and about 100% relative humidity for 28 days.

#### 2.1.2. Bioproducts

The bioproducts, resulting from exploiting of mixed microbial cultures (MMC) grown with crude glycerol as substrate, was produced with a culture medium containing tap water and ammonia as nitrogen source. The PHA-accumulating culture enrichment was performed in a reactor operated as a sequencing batch reactor (SBR) with a working volume of 1500 mL [14] and operated under feast/famine conditions [36]. The type of PHAs produced by the MMC were mainly of the scl-PHA type, P(3HB) or P(3HB-3HV). The waste biomass from the reactor was diluted with water for a concentration of 2–3 g/L and two bioproducts suspensions were obtained: SG (MMC sonicated to breakdown bacterial cell membranes) and NSG (MMC in which the suspension contained whole, non-sonicated, cells). The GP bioproduct was subject to six-cycle of sonication, each with 3 min of ultrasonication followed by a 3 min recycle delay. Both obtained bioproducts yielded a low density, low viscosity and readily applicable formulations to concrete surfaces.

### 2.2. Treatment of Concrete Surface

The surfaces of concrete specimens were treated with bioproducts in an environment at 40 ± 5% relative humidity and 20 ± 2 °C. For the Karsten tube penetration test, fourteen 50 mm × 50 mm × 100 mm specimens were cut from the molded samples produced. The specimens were divided into three groups, of which five specimens were treated with NSG (non-sonicated) bioproduct, another five were treated with the SG (sonicated) bioproduct and the remaining four specimens were treated with tap water (here designated as H_2_O) serving as reference group. The treatments were applied in a 50 mm × 50 mm cut surface that was not in contact with the mold after 28 days of curing. Two coats of treatment, at a concentration of 0.1 mL/cm^2^, were applied drop by drop with pipettes to better ensure uniform distribution across the treated surface and prevent bioproduct overflow.

For the water absorption capillary test, ten cubic specimens of 100 mm × 100 mm × 100 mm were made. The specimens were divided in three groups of three specimens each, except for the reference specimens which were four. All of them were treated in the cut area (that was not in contact with the mold), at an age of 28 days of curing.

### 2.3. Test Method

The tests used to determine the effectiveness of the bioproduct protection to the surface of concrete are described below.

#### 2.3.1. Water Absorption under Low Pressure of Hardened Concrete

The main objective of this test was to analyze the resistance to water penetration [37] after treatment of the concrete surface with the bioproducts. It simulates the pressure of driving rain, a pressure that can “break” the water repellent behavior of a treated material [38]. The test was performed after three days of the application of the bioproducts, with Karsten tubes. Karsten tubes consisted of a 30 mm diameter dome joined with a calibrated glass tube with volumetric graduation (10 mL = 10 cm water column) (Figure 1).

For the test, using the standard EN 16302 [39], the glass dome was adhered to the test surface by means of an airtight seal. In this way it was possible to compare the amount of time it took to absorb 1 mL of water, under the pressure of 10 cm of water column. The reason for limiting the volume of water absorbed by the specimens to 1 mL was due to the high absorption times which occurs naturally in the concrete surfaces. The number of specimens used for each treatment (NSG, SG and H_2_O) allowed the average absorption time to be calculated for each of the biotreatments. To be able to analyze the evolution of the effect of the bioproducts on the treated surface over time, specimens were tested after 3, 7, 14, 21, 28, 42, 60, and 90 days after the biotreatment. 

#### 2.3.2. Water Absorption by Capillarity of Hardened Concrete

To ensure water absorption by capillarity only on the faces treated with the bioproducts, the sides of the specimens were sealed with wax with a 1 cm high layer. The specimens were conditioned according to EN 83966 [40] and the test was performed following EN 83982 [41].

Three days after the application of the different bioproducts, the test was carried out, with humidity and temperature conditions in the test room of 20 ± 2 °C and a relative humidity of 45 ± 15% according to the standard. A lid container with a levelling plastic grille inside (Figure 2) on which the specimens were placed, was used. The specimens were placed on a level grid and in contact with a 5 mm high water layer and were weighted at intervals of: 5 min, 10 min, 15 min, 30 min, 1 h, 2 h, 3 h, 4 h, 6 h, 24 h, 48 h, and 96 h, until the mass was constant (when the difference in mass between two consecutive weighing was less than 0.1%). 

In order to more deeply investigate the efficacy of the biotreatments, the capillary absorption coefficient (*k*) of each of the specimens was calculated, using Equation (1) taken from EN 83982 [34], in combination with the experimental capillary curves:(1)$$k=\frac{\delta a*\varepsilon }{10*\sqrt{m}}$$
where *k* is the capillary absorption coefficient (kg m^−2^ min^0.5^), *δa* is the density of the water considered (1 g/cm^3^), *ε* is the effective porosity of the previously calculated concrete (cm^3^/cm^3^) and m is the water penetration resistance by capillary absorption calculated previously (min/cm^2^).

#### 2.3.3. SEM and EDS Analysis

For the concrete specimens, in addition to the Karsten tube and capillary absorption tests, SEM and EDS analysis were carried out to microscopically analyze the disposition and chemical compositions of the bioproduct on the concrete surface. Tests were performed on a scanning electron microscope type JSM-6980LV (Jeol Ltd., Tokyo, Japan), using AZtec (Aztec SP2, version 4.0; EDS Software of Oxford instruments: High Wycombe, UK, 2018), coupled to EDS-detector type Oxford instrument ultimmax (High Wycombe, UK). Four specimens were analyzed, two for each biotreatment. Two of them were analyzed with EDS before performing the Karsten tube test, the other two were analyzed after completing the Karsten tube test, choosing those specimens that had obtained the best results, not only considering the longer absorption times, but also the homogeneity on the surface of the specimens. The specimens were cut in dimensions of 20 mm diameter and 10 mm high, to perform the SEM and EDS analysis. For SEM data acquisition, a large field detector (LFD) and a pressure of 0.3 mbar were chosen. Images were taken at a magnification of 100×, and an acceleration voltage of 20 kV. For EDS analysis, the same conditions as SEM data acquisition were used.

## 3. Results and Discussion

### 3.1. Water Absorption under Low Pressure

Figure 3 shows the mean absorption times of the different specimens treated with the two bioproducts NSG, SG, and the reference, in which only water was applied. The evolution of the behavior of the different treatments over time is also represented through the eight test results made at different ages.

After three days of treatment and according to the obtained results, both bioproducts increased the water absorption times, by 2.7 times for the SG and 2.3 times for the NSG this result can be primary justified by the hydrophobic nature [26] of PHA and the presence of other organic compounds [42] associated with the MMC [43]. Over time the water absorption capacity of treated concrete specimens decreased progressively, as with the different treatments analyzed by Zhao et al. [44]. This behavior is probably due to the washing effect caused by the Karsten tube test performed successively in the same surface. Nevertheless, after 90 days and eight Karsten tube water absorption tests, the specimens treated with the bioproducts showed an uptake time 30% longer than the reference specimens. The washing effect was also observed in the reference specimens, decreasing the uptake times by more than 14%. Liu et al. [45], studying the microbially induced carbonate precipitation on clay tiles, showed a reduction between 46% and 92% in the water absorption rate.

In the Figure 3, it was also possible to observe the difference in behavior between both bioproducts. During all tests, a higher mean uptake time was detected in the specimens treated with SG, accounting for more than 22% in comparison to the ones treated with NSG. This result could be explained by the nature of the bioproduct since, after the sonication process, the cell walls of the MMC were broken down, releasing the cellular content that acted differently on the concrete surface and presenting a more homogeneous treatment, which was translated into longer absorption times.

### 3.2. Water Absorption by Capillarity

The body of water accumulated as a function of time, showing the capillary absorption process of the specimens, is shown in Figure 4. The entire capillary absorption process could be divided into a rapid first step, dominated by the capillary force when the waterfront had not reached the top of the specimen [46,47] and a second step mainly governed by the diffusing out of the entrapped air after the waterfront reached the top of the specimen [46,48]. The results showed that both treatments had a significant effect when water penetrates by capillary suction, improving by more than 20% the protective capacity for SG and more than 13% for NSG. In comparation, the use of cactus on concrete applied by impregnation on the surface, reduces capillary absorption about 40% Chandra, et al. [49], De Muynck et al. [50] and Achal et al. [51] using induced bacterial carbonate precipitation on the surface of mortars, reduced the capillary water absorption by 40–60% and 84%, respectively. Chandra and Aavik [52] observed a reduction in capillarity between 37.5% and 87.5% when introducing black gram in addition as admixture in concrete. On the other hand, comparing the results with those obtained by Andreotti et al. [53], when using a solution of the terpolymer PHBVV on stone, a reduction on the degree of protection to capillary absorption of 68–91% was observed. Furthermore, comparing the effect of the use of MMC grown with crude glycerol in the formulation of air lime mortars in Oliveira et al. [43], a lower protective capacity of the bioproduct against capillary absorption (17%) was observed. As can be observed, the variability in the results is very large, depending directly on both the characteristics of the substrate [54], on the nature of the treatment used and on the manner of application of this, if kneaded or superficially [49].

All specimens performed consistently from the first weighing to the last one. The reference specimens (treated with water) were the ones that absorbed the highest amount of water, followed by the specimens treated with NSG and, finally, the specimens treated with the sonicated bioproduct were the ones that absorbed the lowest amount of capillary water. As seen in the Figure 4, SG was more effective in the surface protection of concrete than NSG, reducing the absorption of water by capillarity by up to 10%, which was in line with the results obtained in the Karsten tubes test. The explanation to this efficacy may follow the same reasoning: the rupture by sonication of the MMC cell walls makes the bioproduct more effective.

The efficacy of the bioproducts as hydrophobic agents on the concrete surface, can be further supported by the mean values of capillary absorption coefficients (*k*) shown in Table 3. As can be noted, the values obtained fulfil the minimum requirement for the coated concrete ingress established in EN 1504-02 [55], which the water permeability coefficient should not exceed 1.29 × 10^−2^ kg m^−2^ min^0.5^. Those values confirmed that the use of the bioproducts reduced the absorption coefficient in the case of SG by more than 17% and in the case of NSG by 14.5% when compared to the reference specimens.

### 3.3. SEM and EDS Analysis

The images obtained by SEM analysis allowed to observe the physical barrier created by the bioproducts on the surface of the treated concrete specimens, Figure 5. It was possible to distinguish the lifting that had occurred in some points of the surface due to the effect that the water wash had on the biopolymer treatment as for the SG specimen (Figure 5a). Also, it was possible to observe that the sonicated bioproduct had a distribution more homogeneous that the non-sonicated (Figure 5). On the other hand, in the mapping of carbon element of the two biopolymers after (Figure 6) it is possible to appreciate that the percentage of carbon content after the test with the Karsten tubes is higher in SG than in NSG. 

The EDS spectra (Figure 7) allowed to identify the elemental surface composition of the untreated (control) concrete specimens, and treated with the two bioproducts, before and after eight cycles of tests with the Karsten tubes (Table 4). As can be seen, the carbon content of untreated concrete is 10%, which accounts for 49% in the case of the sonicated biopolymer and 44% in the non-sonicated, due to the organic origin of the biopolymers. After the Karsten tube test, the percentage of carbon content decrease of 48% in the SG specimens and 52% in NSG specimens, caused by washing the surface of the specimens as a result of the test with water under pressure was observed. As can be seen from these values, the observation in Figure 6 is corroborated, the percentage of carbon content in the SG was higher than in NSG, before and after the Karsten tube test, with an approximate difference before the test of 10%, and to 15% after the test. All SEM-EDS results indicated that the sonicated bioproduct when applied as a surface treatment was more resistant to washing, had a higher concentration of intracellular components of MMC due to the rupture of the cell walls and has a more homogeneous distribution on the surface of the concrete. Additionally, it confirmed the results obtained in the two previous tests, where SG was more effective, presenting a more significative water-repellent effect than NSG, as in García-González et al. [56], with the application of MMC bioproduct on cement mortar, limestone, and brick. 

## 4. Conclusions

Bioproducts obtained from waste biomass of PHA-accumulating MMC using crude glycerol as a carbon source, either sonicated and non-sonicated, were applied on the cut surface of concrete produced with recycled aggregates. The treated surfaces were tested for water absorption by capillary and under low pressure. The treated surface was observed by SEM-EDS.

Based on the results obtained in this research, the following conclusions can be drawn:The bioproduct subjected to the sonication process, a process in which the cell walls of the MMC were broken, showed a higher effectiveness than the non-sonicated bioproduct; andWith the assessment of the behavior of the biotreated concrete specimens over 8 cycles of Karsten tube tests, it was shown that, although the effectiveness of the biotreatment decreases over time, they still showed a high resistance to water absorption when compared to the reference specimens (treated with water), presuming a longer useful life of recycled concrete.

Furthermore, regarding environmental impact, the incorporation of crude glycerol as the carbon source for the production of the bioproduct used to improve the capacities of recycled concrete and its durability is perceived as a promising solution to reduce the environmental impact of both construction waste and biodiesel by-product, in terms of pollution, waste disposal, energy consumption, and natural resources. 

In summary, the present study demonstrated the effectiveness of these bioproducts to protect the surface of recycled aggregate concrete, preventing its deterioration caused by penetration of water and other possible harmful external agents. Being a potential way of improving the performance of recycled concrete, it can contribute to a more spread use of recycled C&DW aggregates, thereby helping to reduce the volume of such waste.

## Figures and Tables

**Figure 1 materials-14-02057-f001:**
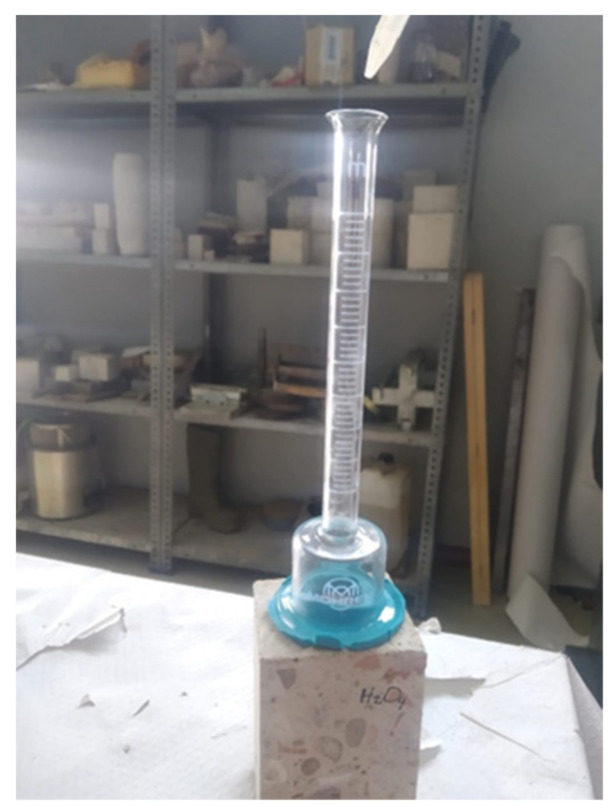
Concrete specimens tested by Karsten tube absorption test.

**Figure 2 materials-14-02057-f002:**
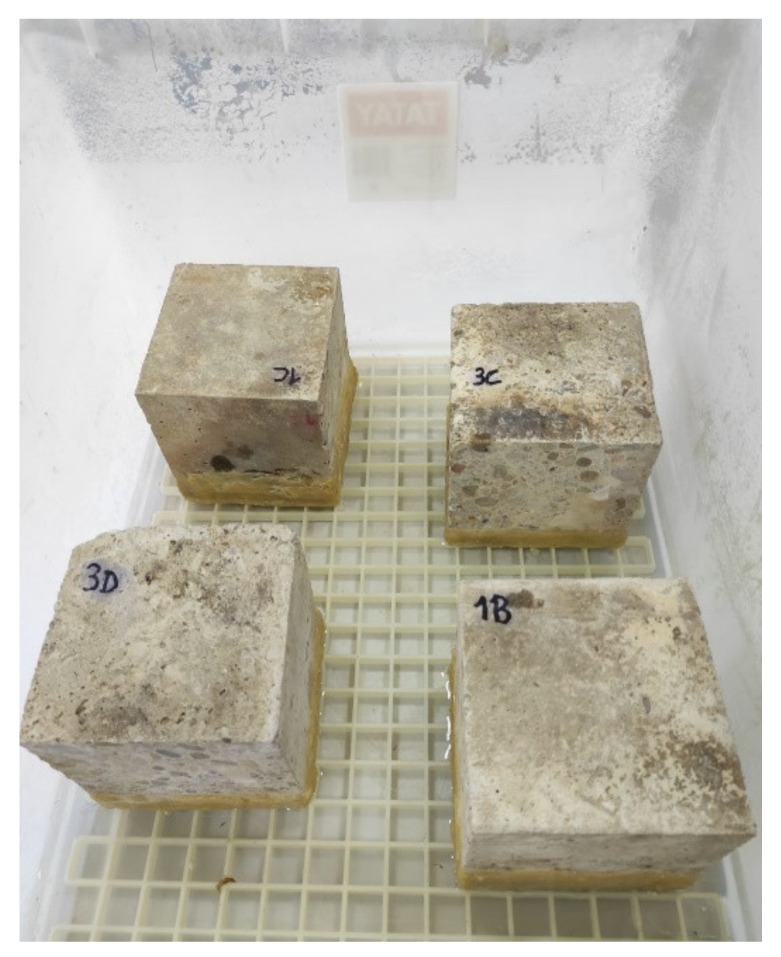
Concrete specimens in the capillary absorption test.

**Figure 3 materials-14-02057-f003:**
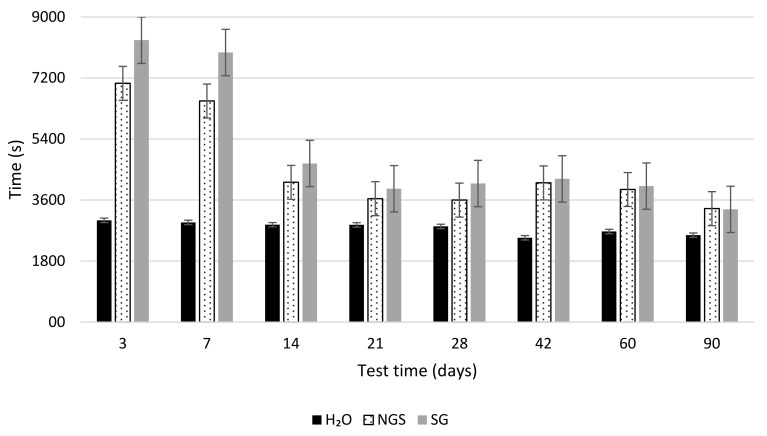
Result of Karsten tube water absorption test for NSG, SG, and H_2_0 treated concrete specimens.

**Figure 4 materials-14-02057-f004:**
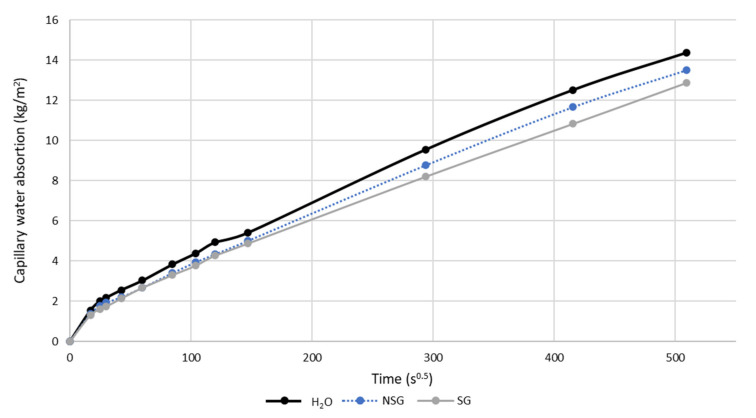
Result of water absorbed by capillarity test on concrete, treated with non-sonicated glycerol (NSG), sonicated glycerol (SG), and water.

**Figure 5 materials-14-02057-f005:**
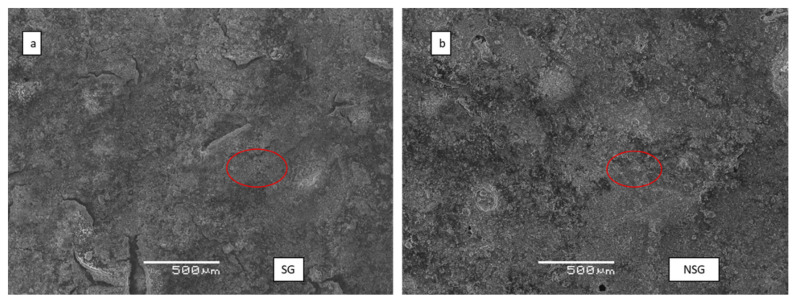
SEM images of SG (**a**) and NSG (**b**) after Karsten tube test. The red circles indicate the point from which the EDS spectra was obtained after the Karsten tube test.

**Figure 6 materials-14-02057-f006:**
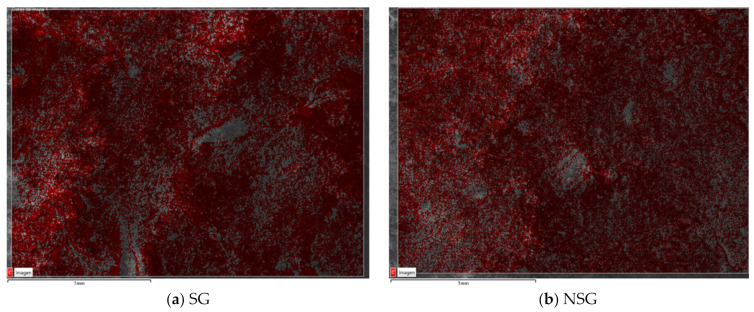
Secondary electron image of mapping carbon element of SG (**a**) and NSG (**b**) after Karsten tube test.

**Figure 7 materials-14-02057-f007:**
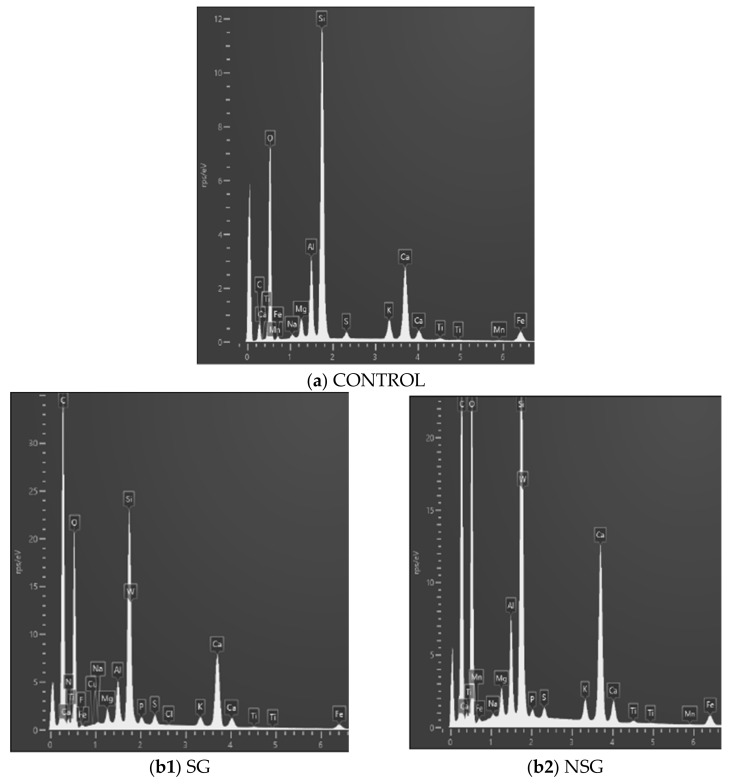
EDS spectra of control (untreated) (**a**), biopolymers before performing the Karsten tube test (**b**) and after the Karsten tube test (**c**).

**Table 1 materials-14-02057-t001:** Recycled concrete composition per m^3^.

Materials	Composition (m^3^)
Water (L)	215
Cement (kg)	391
Sand (kg)	716
Gravel (kg)	447
MRA (kg)	447

**Table 2 materials-14-02057-t002:** Mixed recycled aggregate components.

Components of MRA	% (wt)
Unbound aggregate (natural aggregate with no attached cement mortar)	17.5
Masonry and fired clay (bricks, tiles, stoneware, sanitary ware…)	33.6
Concrete and mortar (natural aggregate with bound cement mortar)	44.1
Asphalt	0.4
Glass	0.8
Gypsum	3.5
Other impurities (wood, paper, metals, plastic…)	0.1

**Table 3 materials-14-02057-t003:** The measurement result of capillary water absorption coefficient of concrete specimens with recycled aggregate treated with NSG, SG, and water.

Treatments	Capillary Water Absorption Coefficient (kg m^−2^ min^0.5^)	Standard Deviation
H_2_O	4.57 × 10^−4^	4.05 × 10^−5^
NSG	3.91 × 10^−4^	3.84 × 10^−5^
SG	3.78 × 10^−4^	3.57 × 10^−5^

**Table 4 materials-14-02057-t004:** Chemical compositions of recycled concrete without any treatment (control) and with NSG and SG biofilms before and after tubes Karsten test.

	Elemental Relative Content (%)				
**Specimens**	C	O	Ca	Si	Al
**Control**	10.0	47.0	8.90	20.80	5.3
**NSG_before test_**	44.3	36.1	6.70	7.80	2.0
**NSG_after test_**	21.5	46.1	19.3	9.30	1.4
**SG_before test_**	49.0	33.1	4.10	6.20	1.3
**SG_after test_**	25.4	45.3	12.5	12.6	1.5

## Data Availability

The data presented in this study are available on request from the corresponding author.

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
