# Peer review of "Use of Bioproducts Derived from Mixed Microbial Cultures Grown with Crude Glycerol to Protect Recycled Concrete Surfaces"

_materials, 2021, doi:10.3390/ma14082057_

Round 1

Reviewer 1 Report

Please find my comments in the attached file

Author Response

Point 1. From the presented SEM micrographs, the deposition of the bioproducts’ layer on the concrete surface cannot be seen. SEM images mapping the carbon element or at least, micrographs of higher magnification should be used to solve this issue. Furthermore, the inclusion of the reference sample (without the addition of bioproducts) in the SEM analysis would be useful for the comparison of the products. How many points on the samples surface were investigated for the EDX analysis? Have you coated your samples prior to SEM analysis?

 Thanks for the feedback. To complement the information provided by the SEM images, following the advice, images of mapping carbon element have been added, as well as the EDS spectra of the sample without the application of the biopolymer, together with the relative content of elements. All of this is incorporated in section 3.3 SEM and EDS Analysis.

For adjacent issues, the number of points investigated on the surface of the samples for the EDS analysis was 4 per sample. In this case the samples are not coated prior to SEM analysis, this step is not necessary with this type of samples and could be a risk that would affect the results of the superficial treatment.

Point 2. Is the sonication process optimized? How was the sonication checked to deliver a product with a successful breakdown of bacteria cell membranes? Which was the applied frequency value for the sonication process?

Yes, the sonication process was previously optimized in previous investigations. We tested several cycles of sonication fixing the power. We observed the cells under the microscope to be sure they were broken. The power/frequency applied in the sonications was 30 kHz and the power was 80% to the total (80 out of 100 W).

Point 3. Have you assessed the durability performance of concrete samples containing the produced bioproducts? If so, have you seen any improvement on the mechanical performance of the specimens?

The tests carried out in the study seek to evaluate the increase of the durability of the concrete with the surface application of the biopolymers protecting the concrete against the penetration of water, due to the close relationship between the deterioration of the concrete and its permeability. This leads not only to the action of water as an agent causing the deterioration, but also to the transport of aggressive substance like sulphate or chloride ions [12]. It should also be consider that this study is an initial analysis, which is intended to be supplemented by tests currently being carried out as well as others planned to be carried out in the future, such as durability tests, porosity, microstructure, etc., with the aim of being able to carry out a complete characterization of the biotreated concrete.  

However, as this is a surface treatment, the main expected behaviour of the biopolymer is to increase the durability of the concrete against external agents, but not to have much influence on its mechanical characteristics, since the application of the biopolymer was superficial.

Point 4. The production process of such bioproducts as well as their application in the concrete technology are economically viable?

Yes, authors consider that the production and application of this type of bioproduct may be economically viable; the production of PHA using mixed microbial communities combined with the utilization of low value substrates is an opportunity to produce PHA cost-effectively (Moita & Lemos, 2012b). These technologies diminish the relatively large expenses for raw substrates and sterilization (Kleerebezem & van Loosdrecht, 2007) and, consequently, avoid part of the operating costs for the global process. In addition to this, it should be noted that the use of glycerol as a carbon source not only reduces the cost of PHA production but also saves on the management of a residual material with high purification costs for other uses.

Reviewer 2 Report

This article used crude glycerol as a carbon source to cultivate polyhydroxyalkanoates (PHA)-producing mixed microbial cultures. The bioproducts derived from these cultures were applied on the surface of concrete with recycled aggregate to create a protective layer. The water absorption of concrete with recycled aggregates has been reduced by this treatment method. However, the data in this paper is too limited to prove the effectiveness of the treatment method. This article can be improved by considering the following comments:

(1) What are the drawbacks of the concrete with recycled aggregates compare with ordinary concrete? What are the advantages of your research? Please explain it in Introduction.

(2) What mineral components are deposited on the surface of concrete? Does the protective layer affect the strength of the concrete? Does the protective layer dissolved in water? Please give more details.

(3) How does this treatment affect the pore size distribution of the concrete?

(4) For comparison, the SEM image of untreated concrete should be shown in Fig. 5. The higher magnification of the SEM images are needed in this article to give more details about the microstructure. There is no obvious difference between those two SEM images in Fig. 5.

(5) In Fig. 3, why does the difference of water absorption between the untreated and the treated concrete become smaller with the test time?

(6) In Fig. 4, the capillary water absorption of concrete was not significantly reduced by this treatment? Why? Those data cannot prove the effectiveness of the treatment. Please give more proofs.

(7) The quality of Figs. 3 and 4 is insufficient for publication. Please improve them.

(8) The authors should consider the durability of concrete after treatment.

Please consider re-review according to above mentioned comments. I suggest a major revision of the article.

Author Response

Point 1. What are the drawbacks of the concrete with recycled aggregates compare with ordinary concrete? What are the advantages of your research? Please explain it in Introduction.

Thank you very much for the recommendation. As can be seen below, information has been included in the introduction to clarify such important issues.

The main problem with the use of recycled aggregates is that as numerous studies have shown [5]–[10], these kind of aggregates have lower performance compared to natural aggregate. In fact, they present higher water absorption and porosity, lower density and higher Los Angeles abrasion capacity, which causes a decrease in the properties of the concrete with recycled aggregate. For this reason, it is necessary to enhance the properties and extend the service life of all types of concrete and specially the recycled ones.

The advantages of this study is to evaluate this novel treatment to improve the durability of recycled aggregate concrete, in an environmentally friendly way and also to contribute to the management of two very large wastes worldwide, namely crude glycerol and construction waste.

Point 2. What mineral components are deposited on the surface of concrete? Does the protective layer affect the strength of the concrete? Does the protective layer dissolve in water? Please give more details.

There is no mineral component deposited on the concrete surface. For the non-sonicated bioproducts all microbial cell is involved in the reduction of the permeability. In the case of sonicated bioproducts the different intracellular components (proteins, lipids, DNAs, debris Polyhydroxyalkanoates, (PHA)) together with cell debris is what creates a physical barrier to reduce the permeability of concrete.   
Moreover, as this is a surface treatment, the main expected behaviour of the bioproduct is to increase the durability of the concrete against external agents, but not to have much influence on the mechanical characteristics of the concrete, since the application of the bioproduct was superficial.
Like is described in the manuscript PHA are hydrophobic and thus not soluble in water. The reason for the decrease in this layer is the washing effect observed and reflected in the results and in particular in the section 3.3. SEM and EDS Analysis. Although the effect of repeated exposure to water causes a decrease in the protective layer and, therefore, in the effectiveness of the bioproduct, as can be seen from the results, even after 8 cycles of low-pressure water exposure, it remains effective and improves the durability of the concrete against the reference.

Point 3. How does this treatment affect the pore size distribution of the concrete?

Porosity analysis has not yet been carried out, so no data is available, although in the future the idea is to carry out a MIP (Hg intrusion) analysis on control and biotreated samples to assess this property. But from the results obtained so far, it can be concluded that at least superficially the application of MMC produces at least partial a sealing of the external pores, as water absorption decreases, as can be seen in the results. This may be related to a reduction of said porosity, although at the moment it is not possible to know which pore diameter is the most affected or other associated details.

Point 4. For comparison, the SEM image of untreated concrete should be shown in Fig. 5. The higher magnification of the SEM images are needed in this article to give more details about the microstructure. There is no obvious difference between those two SEM images in Fig. 5.

Thanks for the feedback. SEM image of untreated concrete is not currently available but to complement the information provided by the SEM images, images of mapping carbon element of the treated surface of the concrete with the biopolymers after the Karsten tube tests have been added, as well as the EDS spectrum of the untreated concrete, together with the relative content of elements which is included in table 4. All of this is incorporated in section 3.3 SEM and EDS Analysis.

Point 5. In Fig. 3, why does the difference of water absorption between the untreated and the treated concrete become smaller with the test time?

That happens due to the washing effect of the biopolymer, generated over the test cycles with water under low pressure. The barrier created by the bioproduct was slightly washed when in contact with water; this caused its effect to be reduced and, therefore, the absorption of water increased with each cycle, reducing the differences between the treated samples and the reference ones.

Point 6. In Fig. 4, the capillary water absorption of concrete was not significantly reduced by this treatment? Why? Those data cannot prove the effectiveness of the treatment. Please give more proofs.

 The capillary absorption of water on specimens treated with the bioproducts was 20% for the sonicated bioproduct and 13% for the non-sonicated bioproduct with respect to the reference specimens. These values for organic treatments represent a significant reduction. As can be seen in publications such as Chandra et al [39] in Figure 3 or Oliveira et al [43] in Figure 6, shows a similar behaviour of the specimens treated relatively to the reference.

Point 7. The quality of Figs. 3 and 4 is insufficient for publication. Please improve them.

Thanks for the comment; figures 3 and 4 has been replaced in the manuscripts.

Point 8. The authors should consider the durability of concrete after treatment.

The tests carried out in the manuscript seek to evaluate the increase of the durability of the concrete with the surface application of the bioproducts protecting the concrete against the penetration of water, due to the close relationship between the deterioration of the concrete and its permeability. This leads not only to the action of water as an agent causing the deterioration, but also to the transport of aggressive substance like sulphate or chloride ions [12]. It should also be consider that this study is an initial analysis, which is intended to be supplemented by tests currently being carried out as well as others planned to be carried out in the future, such as durability tests, porosity, microstructure, etc, with the aim of being able to carry out a more detailed characterization of the biotreated concrete. 

Moreover, as this was a surface treatment, the main expected behaviour of the biopolymer was to increase the durability of the concrete against external agents, but not to have much influence on the mechanical characteristics of the concrete as long as the application of the biopolymer is superficial.

Reviewer 3 Report

The manuscript entitled: “Use of bioproducts derived from mixed microbial cultures grown with crude glycerol to protect recycled concrete surfaces” is in line with the Materials journal. It based on original research. The article is well organized, however it requires some changes:

  • Abstract: please add measurable results.
  • Keywords: “;” instead of “,” after first keyword.
  • Introduction (line 74 and 75): please change the font of the brackets - not in upper index.
  • Concrete (line 108): please clarify if any strength test was applied as a part of investigations.
  • Concrete (line 116 and 117): what for two types the samples dedicated for bending and compressing tests are prepared? What king of investigation this samples have been used?
  • Discussion: lack of discussion and comparison achieved results with literature (!).
  • References: please use the punctuation coherent with the journal template.

Author Response

Point 1. Abstract: please add measurable results.

Thank you very much for the idea. the information has been added in the abstract (line 24-26).

Point 2. Introduction (line 74 and 75): please change the font of the brackets - not in upper index.

Thanks for the comment, this mistake has been corrected in the manuscript.

Point 3. Keywords: “;” instead of “,” after first keyword

Thanks for the comment, this mistake has been corrected in the keywords section .

Point 4. Concrete (line 108): please clarify if any strength test was applied as a part of investigations

Thank you for the comment, the values of compression strength are known due to previous studies carried out with the same concrete (dosage and aggregates) by García-González et al. [34].  

Point 5. Concrete (line 116 and 117): what for two types the samples dedicated for bending and compressing tests are prepared? What king of investigation this samples have been used?

Thank you very much for the comment, the authors don´t described samples because no used for testing in the present study. The assays and the samples used in this study were those described in materials and methods; the compression tests were carried out in previous studies (Garcia-González et al. [34]). The present study aims to assess the influence of bioproducts on the durability of recycled aggregate concrete, by reducing its water absorption.

Point 6. Discussion: lack of discussion and comparison achieved results with literature (!).

Thank you very much for the comment. Due to the few studies that exist related to biotreatments on concrete with characteristics similar to those biopolymers, the comparison with other studies was a challenge. The modifications made has been included in discusion section, in subsetions: 3.1.Water Absorption under Low Pressure (line 230-233), 3.2. Water Absorption by Capillarity and (line 260-261), (line 267-269), and (line 281-283) 3.3. SEM and EDS Analysis (line 297-300) and (line 303-311).

Point 7. References: please use the punctuation coherent with the journal template

Thanks for the comment, this mistake has been corrected in the reference section.

Round 2

Reviewer 3 Report

The authors implemented all necessary changes.

Author Response

We sincerely thank the reviewer for all his comments and suggestions, which have allowed us to improve the article in its different sections